# Utility of Simultaneous Biatrial Atrial Anti-Tachycardia Pacing for the Termination of Atrial Fibrillation during Catheter Ablation of Atrial Fibrillation

**DOI:** 10.3390/jcm11030575

**Published:** 2022-01-24

**Authors:** Shingo Maeda, Masahiko Goya, Yasuhiro Shirai, Atsuhiko Yagishita, Susumu Tao, Jackson Jeikai Liang, Ruben Casado Arroyo, Yoshihide Takahashi, Mihoko Kawabata, Tetsuo Sasano, Kenzo Hirao

**Affiliations:** 1Arrhythmia Advanced Therapy Center, AOI Universal Hospital, Kawasaki 210-0822, Japan; mihoko_kawabata@hotmail.com (M.K.); kenzohirao0926@gmail.com (K.H.); 2Heart Rhythm Center, Department of Cardiovascular Medicine, Tokyo Medical and Dental University, Tokyo 113-8510, Japan; cameister58.cvm@tmd.ac.jp (M.G.); sumtao1978@yahoo.co.jp (S.T.); sasano.cvm@tmd.ac.jp (T.S.); 3Heart Rhythm Center, National Hospital Organization Disaster Medical Center, Tokyo 190-0014, Japan; whity_yasuo@yahoo.co.jp; 4Department of Cardiology, Tokai University School of Medicine, Isehara 259-1193, Japan; a.yagisita@gmail.com; 5Department of Cardiac Electrophysiology, University of Michigan, Ann Arbor, MI 48109, USA; liangjac@med.umich.edu; 6Department of Cardiology, Hôpital Erasme, Université Libre de Bruxelles, 1070 Brussels, Belgium; ruben.casado.arroyo@erasme.ulb.ac.be; 7Department of Cardiology, Shinyurigaoka General Hospital, Kawasaki 215-0026, Japan; yoshihide_takahashi@oboe.ocn.ne.jp

**Keywords:** atrial fibrillation, atrial anti-tachycardia pacing, catheter ablation, continuous wavelet transform, coefficient of variation

## Abstract

Background: Atrial anti-tachycardia pacing (A-ATP) of the right atrium (RA) has been shown to decrease the burden of atrial fibrillation (AF) in patients with dual-chamber pacemakers. The aim of this study is to identify the novel predictors of effective A-ATP for terminating AF in patients with AF undergoing catheter ablation. Methods: This study included 41 consecutive patients undergoing a first ablation procedure for paroxysmal (PAF: *n* = 21) or persistent (PEF: *n* = 20) AF. We prospectively evaluated predictors of AF termination after A-ATP. The coefficient of variation (CoV = SD/mean × 100) of the dominant frequencies (DFs) was calculated to evaluate the variability in atrial activation. Results: AF was terminated by A-ATP in 29% of PAF and 5% of PEF patients. In these patients, simultaneous high-rate pacing from the RA and the coronary sinus (CS) terminated AF in 71% of patients, in whom the mean AF cycle length (CL) before A-ATP was longer (214 ± 23 vs. 177 ± 35 ms, *p* = 0.02) and became slower after A-ATP (234 ± 37 vs. 176 ± 32 ms, *p* < 0.01), compared to unsuccessful patients. The CoV of the DFs before A-ATP were lower in both RA (6.2 ± 2.0 vs. 15.3 ± 7.9, *p* = 0.02) and CS (11.0 ± 7.9 vs. 24.3 ± 9.3, *p* < 0.01) in successful patients. Conclusions: Simultaneous biatrial A-ATP from the RA and CS could terminate AF in patients with PAF. The predictors for successful termination include longer AF CL and higher AF stability.

## 1. Introduction

Atrial fibrillation (AF) is a common arrhythmia that can result in symptoms, heart failure, tachycardia-mediated cardiomyopathy, and most importantly, stroke and systemic embolism. Increased device-detected AF burden and duration appear to be associated with increased risk of cerebral ischemic events in patients with implanted devices [1]. The ‘MINimizE Right Ventricular pacing to prevent Atrial fibrillation and heart failure’ (MINERVA) multicenter randomized study described atrial anti-tachycardia pacing (A-ATP) has been shown to decrease the burden of AF in patients with bradycardia and atrial tachyarrhythmias [2]. From this study, we know AF can be terminated by A-ATP; however, the mechanism of AF termination is still unknown. Furthermore, there are no studies exploring why AF can be terminated by A-ATP. In this study, we sought to evaluate the novel predictors of effective A-ATP for terminating AF in patients with AF undergoing catheter ablation.

## 2. Materials and Methods

### 2.1. Patient Population

This study enrolled 41 consecutive patients referred to the Tokyo Medical and Dental University undergoing first radiofrequency catheter ablation (RFCA) of AF. Paroxysmal AF (PAF) is defined as AF that has been cardioverted during the first 7 days. Persistent AF (PeAF) is defined as AF that persists without interruption for 7 days or longer [3,4]. All data were collected in a registry database approved by the Institutional Review Board of the Tokyo Medical and Dental University (#M2018-062). Waiver of consent was obtained through the institutional review board for retrospective review of the imaging and clinical data.

### 2.2. AF Ablation Protocol

All patients underwent a transesophageal echocardiogram before ablation to exclude left atrial thrombus. Warfarin was continued, and direct oral anticoagulant was interrupted just one dose before the procedure. The procedure was performed under deep sedation with dexmedetomidine. Four venous access sites were obtained in the right femoral vein. Heparin was administrated to maintain an activated clotting time of 300–350 s during the procedure. A 20-pole catheter with intracardiac defibrillation capability was inserted into the coronary sinus (CS). After transseptal puncture, a 20-pole circumferential mapping catheter through the SL0 sheath (Abbott, Minneapolis, MN, USA) and a 3.5-millimeter open-irrigated-tip ablation catheter (ThermoCool SmartToch^TM^ SF, Biosense Webster Inc., Diamond Bar, CA, USA or TactiCath^TM^, Abbott, Minneapolis, MN, USA) through a steerable sheath (Agilis; Abbott) were introduced to the left atrium (LA). Catheter ablation was performed under the guidance of a three-dimensional electroanatomic mapping (EAM) system (CARTO, Biosense Webster; or Ensite NavX, Abbott, Minneapolis, MN, USA). A high-resolution EAM of LA was created with the PENTARAY^TM^ (Biosense Webster) or an Advisor™ HD Grid (Abbott, Minneapolis, MN, USA) mapping catheter and RFCA lesions were given for targeting of ablation index 450 or lesion size index 5.0 for pulmonary vein isolation (PVI). The endpoint of the PVI procedure was achieving entrance block, defined as the absence of the local PV potentials, and exit block.

### 2.3. Pacing Protocol

During AF, decremental pacing and/or straight pacing were attempted at outputs of 10 V with 1 ms pulse width from the high right atrium (HRA), CS, and simultaneously from both the HRA and CS until termination of AF. If patient’s rhythm was sinus rhythm at the time of procedure, we induced AF by burst pacing from CS (pacing cycle length (CL): 200 ms) on isoproterenol (1–2γ). After AF sustained for more than 30 s, we attempted pacing maneuvers. We defined successful AF termination as a termination within 10 s after pacing with obvious CL changes. Outcome after A-ATP were classified into 3 results: No termination, direct termination, and delayed termination.

The pacing protocol used for this study is as follows (Figure 1):

First, we measured mean AF CL for 10 beats. Second, we attempted pacing from HRA using below five pacing CL until AF was terminated (Th1→Th2→Th3→Th4→Th5). If AF was not terminated, we changed pacing site to CS distal. If AF was not terminated again, we finally performed simultaneous HRA and CS distal pacing. Pacing study was finished when AF was terminated. Percentage of pacing CL was referred to the protocol of the MINERVA study [2] as follows:A.Th1. Decremental pacing: 91% of AF CL × 13 beats, reduce by 10 ms (down to 150 ms)B.Th2. Straight pacing: 84% of AF CL × 13 beats, reduce by 10 ms (down to 150 ms)C.Th3. Decremental pacing: 81% of AF CL × 13 beats, reduce by 10 ms (down to 150 ms)D.Th4. Straight pacing: 84% of AF CL × 20 beats, reduce by 10 ms (down to 150 ms)E.Th5. Decremental pacing: 81% of AF CL × 20 beats, reduce by 10 ms (down to 150 ms)

Decremental pacing in this study means “Pacing CL was reduced by 10 ms after each 13/20 pacing beats till 150 ms”. (ex; AF CL = 200 ms, Th1; 180 ms × 13 beats + 170 ms × 13 beats + 160 ms × 13 beats + 150 ms × 13 beats).

### 2.4. Electrocardiogram Acquisition

Endocardial filtered (30 to 500 Hz) bipolar electrocardiograms were recorded using a multi-electrode catheter at the sampling rate of 1000 Hz during atrial fibrillation.

### 2.5. Continuous Wavelet Transform (CWT) Analysis

Acquired electrocardiograms were analyzed offline during the ablation procedure using self-developed CWT analysis software [5,6]. This software was developed using free software “R” (https://www.r-project.org (accessed on 10 January 2020)) in volunteered cooperation with Nihon Koden Corporation, Tokyo, Japan. Butterworth filter 10 to 50 Hz was used for processing the raw data to the software. The CWT is defined as the sum overall time of the signal multiplied by scaled (frequency) and shifted (time) versions of the adopted mother wavelet. Morlet wavelet was used as the mother wavelet in this study. The waveform transformed into a scale, then translated to a pseudo-frequency (PF). The coefficient of variation (CoV) was defined as (standard deviation of 3000 PF data)/(arithmetic mean), of which <20 was arbitrary judged as a spatiotemporally stable wave signal.

### 2.6. Statistical Analysis

Statistical analyses were performed using SPSS version 21.0 software (SPSS Inc, Chicago, IL, USA). A two-tailed *p* value < 0.05 was considered statistically significant. Continuous data are presented as mean and standard deviation or median and interquartile range when skewed. Categorical data are presented as percentages. Comparisons between groups were made using two-sample T test or one-way ANOVA.

## 3. Results

### 3.1. Patient Characteristics

Of the 41 patients undergoing ablation for AF (mean age 63 ± 12 years, 93% male gender), there were 21 (51.2%) patients with PAF and 20 (48.8%) with PeAF. In patients with PAF, fourteen patients presented in sinus rhythm, whereas no patients presented in sinus rhythm at the time of the procedure. All patients had normal left ventricular ejection fraction and no evidence of structural heart disease. There were no significant differences in age, gender, comorbid conditions, CHADS_2_ score, medication usage, and left ventricular ejection fraction between the two groups (Table 1). Left atrial diameter was significantly larger in patients with PeAF compared to those with PAF (PAF vs. PeAF = 40 ± 5 vs. 46 ± 7 mm, *p* = 0.02). The patients had a mean of 2 ± 1 antiarrhythmic medication, including beta-blocker. Twenty patients were taking anti-arrhythmic drugs (Pilsicainide, *n* = 6; Flecainide, *n* = 7; Bepridil, *n* = 7), and/or twenty-one patients were taking beta-blocker (Bisoprolol, *n* = 21). All medications were stopped before the procedure—at least 7 days.

### 3.2. Representative Cases

#### 3.2.1. Case 1: 42-Year-Old Man with Symptomatic PAF

AF occurred spontaneously before the ablation procedure. Burst pacing was attempted from the RA appendage (RAA) and AF was terminated after 4.5 s (Figure 2).

#### 3.2.2. Case 2: 72-Year-Old Man with Symptomatic PAF

AF occurred spontaneously during mapping. Simultaneous biatrial high-rate straight pacing was delivered from both the CS and RA (Figure 3). AF organized into a more stable tachycardia after pacing and mean AF CL became slower and then terminated. Continuous wavelet transform (CWT) analysis also showed increased AF stability after pacing.

#### 3.2.3. Case 3: 67-Year-Old Man with Symptomatic PAF

AF resulted in organized tachycardia after simultaneous A-ATP from CS and RA and was terminated by decremental pacing (Figure 4). In the CWT analysis, AF became more stable after A-ATP, similar to case two.

### 3.3. Results of A-ATP

AF was successfully terminated with A-ATP in 7 (17%) patients. The rate of successful termination with A-ATP was significantly higher in patients with PAF compared versus PeAF (29% vs. 5%, *p* < 0.05). In patients with successful AF termination, simultaneous HRA and CS pacing was more effective compared to single-site pacing form HRA and CS [Success rate, HRA: 1 pt (14%), CS: 1 pt (14%), HRA and CS: 5 pt (71%)], and decremental pacing (A, Th1) was more effective than straight pacing [Successful decremental pacing: 5 pt (A, Th1; *n* = 3, C, Th3; *n* = 1, E, Th5; *n* = 1), Successful straight pacing: 2 pt (B, Th2; *n* = 1, D, Th4; *n* = 1)]. Since two patients in whom A-ATP from the HRA and CS pacing was successfully terminated AF, they did not try simultaneous A-ATP from CS and RA.

### 3.4. Mean AF CL

In patients with successful AF termination by A-ATP, mean AF CL at baseline was significantly longer compared to patients in whom A-ATP failed to terminate AF (214 ± 23 vs. 177 ± 35 ms, *p* = 0.02). Moreover, in those with successful AF termination by A-ATP, the AF CL became significantly longer after A-ATP (234 ± 37 vs. 176 ± 32 ms, *p* < 0.01)(Figure 5).

### 3.5. CoV of the DFs

In patients with successful AF termination by A-ATP, baseline mean CoV were significantly lower compared to failed patients (AF no termination) at both the CS (11.0 ± 7.9 vs. 24.3 ± 9.3 ms, *p* = 0.02) and RA (6.2 ± 2.0 vs. 15.3 ± 7.9 ms, *p* < 0.01)(Figure 6).

## 4. Discussion

We investigated the predictors of effective A-ATP for terminating AF in patients with AF undergoing catheter ablation. Simultaneous A-ATP from both the RA and CS was able to terminate AF in 29% of patients with PAF compared with only 5% of those with PeAF. In addition, most patterns of AF termination were delayed termination, and decremental pacing appeared to be more effective than straight pacing. In cases of successful termination of AF, atrial capture was seen during A-ATP, which may have contributed to AF termination, as has been previously described for atrial flutter (AFL) [7]. Typical AFL can be easily entrained and terminated with overdrive pacing from the HRA, however, the mechanism of AF termination is still unknown.

### 4.1. Mechanism of AF Termination

Figure 7 shows a representative case involving a 60-year-old man with symptomatic PAF, which may hypothesize one mechanism of AF termination. During simultaneous biatrial straight pacing from the CS and HRA, 1/20 pacing impulses captured the HRA and 4/20 pacing impulses captured the CS (right blue arrows). Because the *p* wave appears very small and narrow and the number of captured pacing beats were small, the effects of pacing from CS and HRA were unclear from 12 leads electrograms. However, the atrial activation sequences across both the HRA and CS catheters became reversed after the pacing maneuvers; at both CS and HRA catheters, the activation sequence changed from “distal to proximal” to “proximal to distal” after pacing (red arrows) before termination of AF.

While the exact mechanism of AF termination remains unclear, three possible hypotheses are proposed (Figure 8 and Figure 9) [8]: (A)Multi-wavelet reentry theory [9,10,11]; Pacing stimulations collide the dominant circuit of a small reentrant circuit, and the balance of a multi-reentry collapses. The AF becomes unstable and cannot sustain, resulting in eventual termination.(B)Mother Rotor theory [12,13]; Pacing stimuli block a main pathway of the mother rotor, preventing it from sustaining, resulting in eventual termination.(C)Leading circle theory [14]; The presence of an excitable gap is well known for the maintenance of AF. Pacing stimuli entrain and fill the excitable gaps of tachycardia, affecting the refractory period. Eventually, AF is unable to sustain itself and thus terminates slowly.

**Figure 8 jcm-11-00575-f008:**
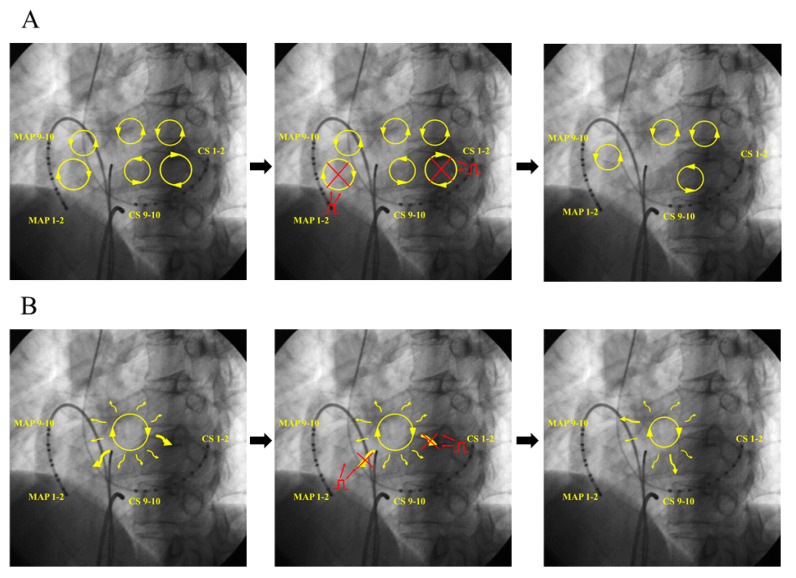
Three possible hypotheses for the mechanisms of AF termination. (**A**) Multi-wavelet reentry theory; Pacing stimulations collide the dominant circuit of a small reentrant circuit, and the balance of a multi-reentry collapses. The AF becomes unstable and cannot sustain, resulting in eventual termination. (**B**) Mother Rotor theory; Pacing stimuli block a main pathway of the mother rotor, preventing it from sustaining, resulting in eventual termination. CS = coronary sinus.

**Figure 9 jcm-11-00575-f009:**
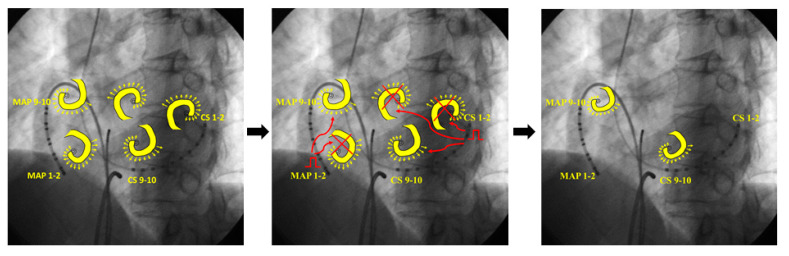
Three possible hypotheses for the mechanisms of AF termination. Leading circle theory; The presence of an excitable gap is well known for maintenance of AF. Pacing stimuli entrain and fill the excitable gaps of tachycardia, affecting the refractory period. Eventually, AF is unable to sustain itself and thus terminates slowly. CS = coronary sinus.

As shown in Figure 7, atrial capture from the pacing stimuli affects AF and can lead to AF termination. Interestingly, our cases demonstrate that the majority of AF termination episodes were delayed rather than direct terminations. If AF were organized (similar to atrial tachycardia), pacing could terminate AF directly. We speculate that the pattern of delayed termination may indicate that when pacing stimuli entrain the main AF circuit (via the aforementioned mechanisms), AF will become unstable and the power of maintenance will become weak. For these reasons, AF will eventually terminate. Therefore, simultaneous biatrial A-ATP from both the HRA and CS can cover the entirety of both atria, and continuous pacing will have a higher likelihood of entraining the AF circuit compared to single-site pacing. Furthermore, since refractory periods and excitable gaps are always changing, decremental pacing may be more effective than burst pacing to adapt to variable AF CLs.

### 4.2. Clinical Implications

This study demonstrates the utility of simultaneous biatrial A-ATP from the CS and HRA for the termination of AF during RFCA of AF. Electric cardioversion to restore sinus rhythm is frequently performed during AF ablation procedures for a number of reasons—improves hemodynamics, accurate voltage mapping, assessing for entrance and exit block after PVI, assessing for bidirectional block across linear lesion sets, non-PV trigger provocation, etc. In some situations (as with mapping for non-PV triggers), multiple cardioversions may be required. The need to perform multiple electrical cardioversions may result in patient motion with a resulting map shift of the electroanatomic map, and can also increase the risk of complications including skin burns, catheter perforation and pericardial effusion, and myocardial stunning. As such, in cases where multiple restorations of sinus rhythm may be necessary, A-ATP can be attempted to restore sinus rhythm before cardioversion to recover sinus rhythm.

The MINERVA multicenter randomized study showed that A-ATP was effective in reducing AF burden in patients with implanted devices [2]. However, the currently available dual chamber and biventricular pacing systems are unable to be programmed to simultaneously deliver biatrial A-ATP from both the HRA and the CS. A supplementary Video shows the propagation of pacing from RAA (video one) and RA septum (video two). During RAA pacing activation sequence advanced to anterior LA via Bachman bundle and spread to posterior LA from RAA pacing, whereas the activation sequence spreads to the anterior and posterior LA simultaneously from the RA septum during pacing. For these reasons, A-ATP delivered from a pacemaker can affect AF, possibly resulting in AF termination. Furthermore, these videos suggest that a septal RA lead location may be more effective than a lateral (RAA) lead location, as propagation from the RA septum during A-ATP conducts anteriorly and posteriorly simultaneously, possibly increasing the likelihood of AF termination.

### 4.3. Study Limitations

This study is a single-center study with a limited sample size. Larger multicenter studies are necessary to confirm the mechanisms and utility of simultaneous A-ATP for AF termination. In addition, the use of new mapping systems CARTO-Finder (Biosense Webster, Inc., Diamond Bar, CA, USA) and ExTRa Mapping™ (Nihon Kohden Co., Tokyo, Japan) may be helpful to delineate the mechanisms of AF termination.

In addition, a major limitation of this study is that the single-site and multi-site pacing was performed in sequence. That is, all patients underwent single site first; then, if that failed, they underwent multi-site pacing. We also know that AF can spontaneously initiate and terminate during an AF ablation regardless of pacing. Therefore, a better study would compare two groups of patients—those who had only a single site vs. those who had only multi-site—or some sort of cross-over where single was done in the first half and multi was done in the other half. Finally, the largest major limitation in this study is the lack of a control group—many patients would have AF terminated spontaneously despite no pacing. Further study is needed to provide some information about what CL and CWT analysis shows in that control group and the rate of spontaneous termination of AF in those who do not get atrial pacing.

## 5. Conclusions

This study is the first report to evaluate the predictors of effective A-ATP for terminating AF in patients with AF undergoing catheter ablation. Simultaneous biatrial A-ATP from the HRA and CS can terminate AF, especially in patients with PAF. Characteristics of AF which predict successful termination by A-ATP include longer AF CL and increased AF stability.

## Figures and Tables

**Figure 1 jcm-11-00575-f001:**
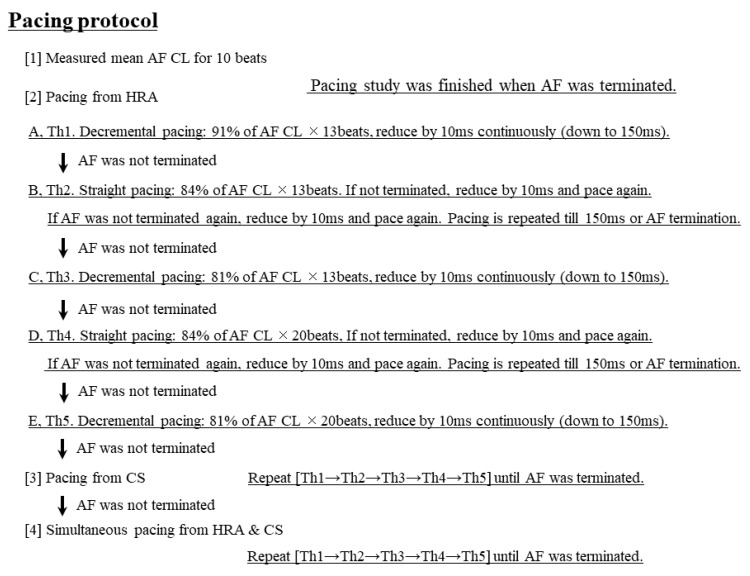
Pacing protocol of this study. AF = atrial fibrillation; CS = coronary sinus; CL = cycle length; HRA = high right atrium; Th = therapy.

**Figure 2 jcm-11-00575-f002:**
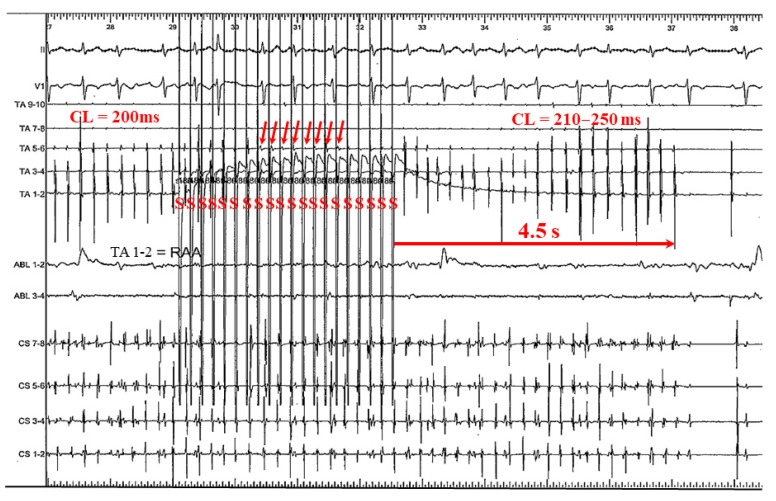
Case 1. Burst pacing was attempted from the right atrial appendage, and AF was terminated after 4.5 s. In this case, 8/20 pacing impulses captured the RA (arrows), resulting in slowing of AF CL prior to AF termination. AF = atrial fibrillation; CL = cycle length; CS = coronary sinus; RAA = right atrial appendage; PAF = paroxysmal AF; S = stimulation: TA = tricuspid annulus.

**Figure 3 jcm-11-00575-f003:**
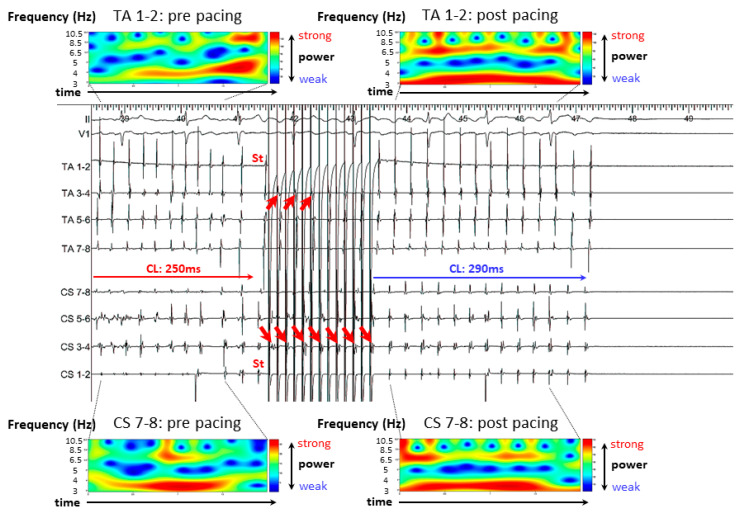
Case 2. AF was terminated by simultaneous biatrial high-rate straight pacing from CS and HRA. Pacing captured the CS or HRA on alternately beats (red arrows), and AF organized into a more stable tachycardia. In the continuous wavelet transform analysis, red lines prolonged after pacing. The most powerful pseudo frequency (= mean dominant frequency) is colored red. AF = atrial fibrillation; CL = cycle length; CS = coronary sinus; HRA = high right atrium; PAF = paroxysmal AF; St = stimulation; TA = tricuspid annulus.

**Figure 4 jcm-11-00575-f004:**
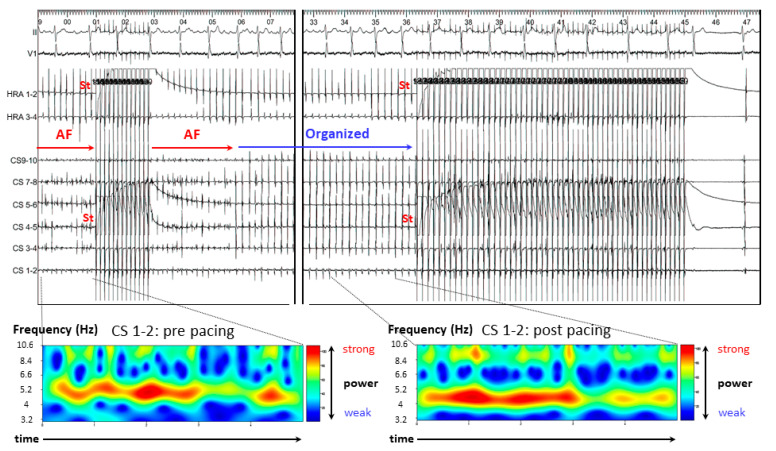
Case 3. AF resulted in organized tachycardia after simultaneous A-ATP from CS and HRA and terminated directly by decremental pacing. In the continuous wavelet transform analysis, red line prolonged after A-ATP. AF = atrial fibrillation; CS = coronary sinus; PAF = paroxysmal AF; HRA = high right atrium; St = stimulation.

**Figure 5 jcm-11-00575-f005:**
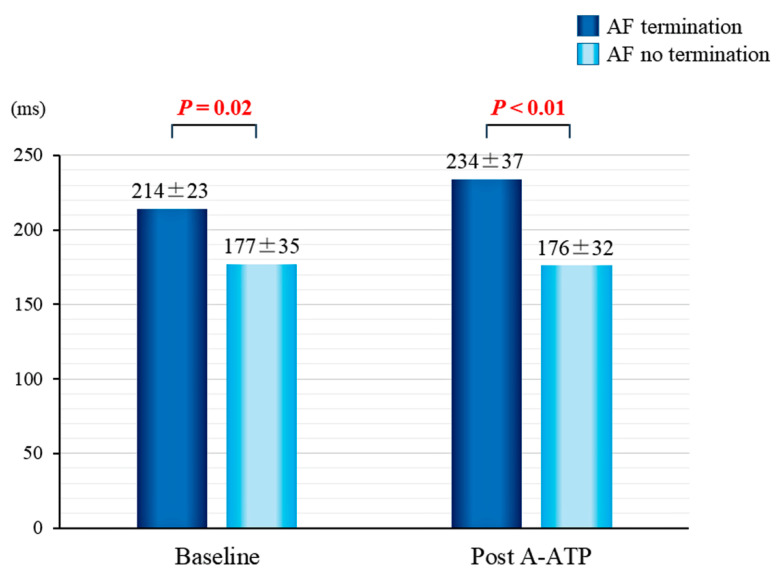
Mean AF CL at baseline and post A-ATP. AF = atrial fibrillation; A-ATP = atrial anti-tachycardia pacing; CL = cycle length.

**Figure 6 jcm-11-00575-f006:**
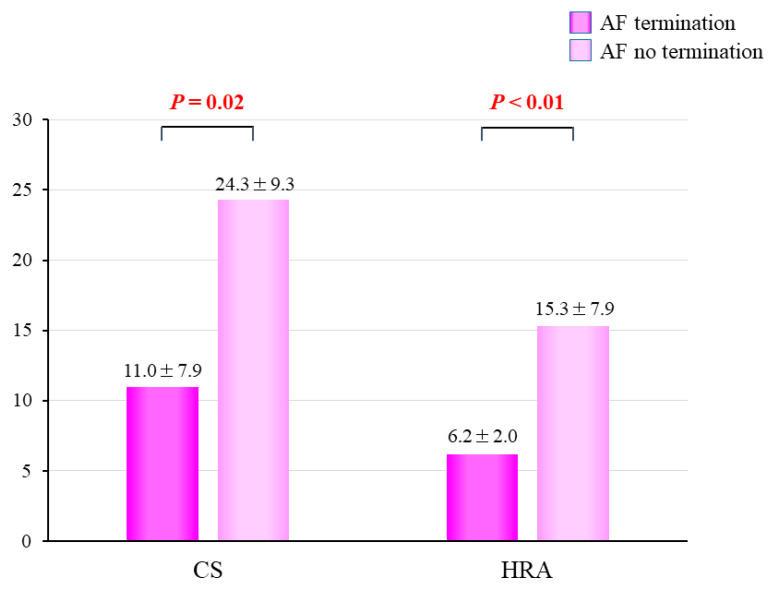
Baseline mean coefficient of variation at CS and HRA. AF = atrial fibrillation; Cs = coronary sinus; HRA = high right atrium.

**Figure 7 jcm-11-00575-f007:**
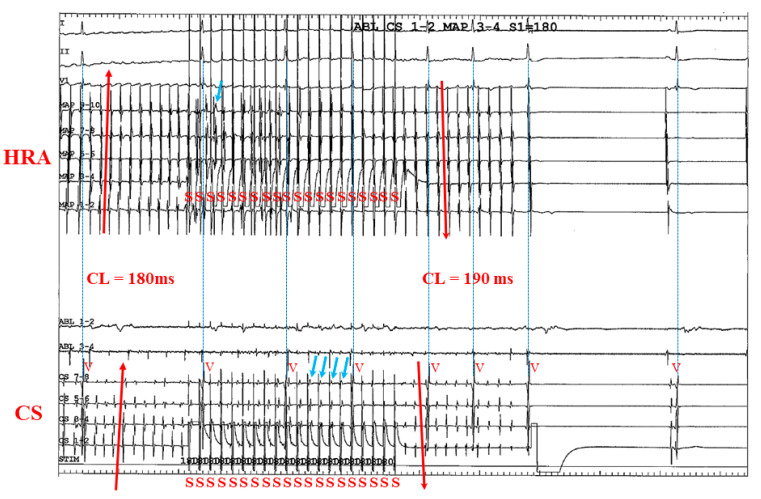
A representative case involving a 60-year-old man with symptomatic PAF undergoing ablation which hypothesizes one mechanism of AF termination. During simultaneous biatrial straight pacing from the CS and HRA, 1/20 pacing impulses captured the HRA and 4/20 pacing impulses captured the CS (right blue arrows). However, the atrial activation sequences across both the HRA and CS catheters became reversed: At both CS and HRA catheters, the activation sequence changed from “distal to proximal” to “proximal to distal” after A-ATP (red arrows) before termination of AF. AF = atrial fibrillation; CL = cycle length; CS = coronary sinus; HRA = high right atrium; PAF = paroxysmal AF; S = stimulation.

**Table 1 jcm-11-00575-t001:** Baseline characteristics (*n* = 41).

	PAF (*n* = 21)	PEF (*n* = 20)	*p* Value
Age	65 ± 12	60 ± 11	0.26
Male, *n* (%)	19 (90)	15 (75)	0.19
Diabetes mellitus, *n* (%)	2 (14)	6 (30)	0.10
Hypertension, *n* (%)	13 (62)	8 (40)	0.16
Congestive heart failure, *n* (%)	0 (0)	3 (15)	0.07
Stroke, *n* (%)	3 (14)	3 (15)	0.95
Age > 75 y.o., *n* (%)	5 (24)	1 (5)	0.09
Chronic kidney disease, *n* (%)	3 (14)	3 (15)	0.95
Coronary artery disease, *n* (%)	4 (19)	3 (15)	0.73
CHADS_2_ score	1.3 ± 1.0	1.3 ± 1.4	0.90
Beta-blocker	8	13	0.08
Anti-arrhythmic drug	13	7	0.08
Pilsicainide	5	1	0.09
Flecainide	5	2	0.24
Bepridil	3	4	0.63
LA diameter, (mm)	40 ± 5	46 ± 7	0.02
Ejection Fraction, (%)	65 ± 6	63 ± 6	0.57

Values are presented as mean ± SD. A *p* value < 0.05 was considered significant. LA = left atrium.

## Data Availability

The data presented in this study are available on request from the corresponding authors.

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
