# Peer review of "Utility of Simultaneous Biatrial Atrial Anti-Tachycardia Pacing for the Termination of Atrial Fibrillation during Catheter Ablation of Atrial Fibrillation"

_jcm, 2022, doi:10.3390/jcm11030575_

Round 1

Reviewer 1 Report

Re: jcm-1532950

Dear Editors,

I have been asked to review the above referenced manuscript. The study is a single center, retrospective study examining 41 consecutive patients undergoing AF ablation. The basic study question is whether an algorithm for single site (HRA or distal CS) or multisite (both) pacing during atrial fibrillation can terminate atrial fibrillation.

The manuscript is well written and clearly conducted. The protocol was adopted prospectively for consecutive patients but data collected retrospectively. The main finding is that atrial pacing during AF can increase AF stability, as measured by CWT analysis, and can terminate atrial fibrillation. Below is a line by line commentary followed by summative comments.

Abstract: second sentence is a run-on sentence and should be edited.

Section 2.1: please clarify details of baseline clinical data. How was “PAF” v “PeAF” defined? By ICD10 coding? Over what time period were these patients enrolled? How many PAF patients presented in sinus rhythm and how many PeAF patients presented in sinus rhythm?

Section 2.2 last sentence needs an “and” before “RFCA lesions were given”

Section 2.3 Pacing protocol. This needs some more clarity. Presumably the 20 patients with persistent atrial fibrillation were already in atrial fibrillation. But what was the specific protocol for PAF patients presenting in sinus rhythm? How was AF induced? Was isoproterenol used, and what doses? How long was AF allowed to continue before the pacing maneuvers were attempted? An obvious concern is that PAF can terminate on its own, especially if it was triggered by mapping, and it is difficult to tease out whether pacing actually affected the substrate. Can you provide CWT analysis for a control group in whom no pacing was performed but AF spontaneously terminated? Perhaps slowing of cycle length occurs any time AF terminates, and it has nothing to do with the atrial pacing?

Protocol: Please clarify what the difference between decremental pacing and burst pacing is. They appear to be the same thing. Why does burst pacing have a 10ms decrease? Is that between pacing trains? How many pacing trains were given and was there a pause? What was the order of attempts? A, B, C, D, E? Or some random order? Was each patient given all 5 pacing strategies, or only one of the five? Was pacing performed until local capture was demonstrated (e.g., arrows on Figure 1) or was there some other criteria to stop pacing?

Section 2.6 Statistical Analysis: please indicate if P values were one-tailed or two-tailed.

Section 3.1. This needs more data. What were baseline medications of the study group? Clearly, medications such as class I, II, III, or IV antiarrhythmics can influence the study results. Were medications stopped prior to the study? For how long?

Please also include information about which algorithm (A-E) was used and which one had effect.

Section 4.1: compared to the sinus P wave, the “proximal-distal CS activation” appears to have closely spaced EGMs. The P wave appears very small and narrow. This really has no significance. Is there a CWT to assess here? Also, Figure 6 is confusing. What is the top panel (“RA”) compared to the bottom panel (“CS”)? The text says that there was simultaneous pacing, yet those two tracings appear to have different surface QRSs and appear to be at different time points. Please clarify.

Section 4.2. A major limitation of this study is that the single site and multi site pacing was performed in sequence. That is, all patients underwent single site first, then if that failed, multi site pacing. We also know that AF can spontaneously initiate and terminate during an AF ablation regardless of pacing. Therefore, a better study would compare two groups of patients, those who had only single site v those who had only multi site; or some sort of cross over where single was done first in half and multi was done first in the other half. Please address this limitation.

Section 4.3. There are many more limitations to this study than addressed. See above.

This is a well written and well conducted study that examines atrial pacing during AF that can terminate AF. However, there are a number of minor concerns that should be addressed. More clarity is needed in the protocol for pacing. Figures need to be clarified. The largest major concern is the lack of a control group – many patients would have AF terminated spontaneously despite no pacing. Can the authors provide some information about what CL and CWT analysis shows in that control group? What is the rate of spontaneous termination of AF in those who don’t get atrial pacing?

Author Response

Dear Reviewer 1

We really appreciate your time and effort for the thorough review of our manuscript. Your comments and questions are very valuable and were quite helpful for improving the manuscript. Please note that the edits and figures have been added in the revised manuscript according to your suggestions. The following is our responses to the questions and comments.

Response to the Comments by Reviewer 1:

Point 1: “Abstract: second sentence is a run-on sentence and should be edited.”

Response to Point 1:

Thank you for your important conceptual observation. We have therefore amended the manuscript as instructed by adding the following paragraph at abstract section.

The aim of this study is to identify the novel predictors of effective A-ATP for terminating AF in patients with AF undergoing catheter ablation.”

Point 2: “Section 2.1: please clarify details of baseline clinical data. How was “PAF” v “PeAF” defined? By ICD10 coding? Over what time period were these patients enrolled? How many PAF patients presented in sinus rhythm and how many PeAF patients presented in sinus rhythm?”

Response to Point 2:

Thank you for highlighting these important comments. Paroxismal AF (PAF) is defined as AF that have been cardioverted during the first 7 days. Persistent AF (PeAF) is defined as AF that persists without interruption for 7 days or longer [3,4]. Furthermore, in patients with PAF, fourteen patients presented in sinus rhythm, whereas no patients presented in sinus rhythm at the time of procedure. We have therefore amended the manuscript as instructed by adding the following paragraph and 2 references at “Section 2.1 and 3.1”.

Paroxismal AF (PAF) is defined as AF that have been cardioverted during the first 7 days. Persistent AF (PeAF) is defined as AF that persists without interruption for 7 days or longer [3,4].”

In patients with PAF, fourteen patients presented in sinus rhythm, whereas no patients presented in sinus rhythm at the time of procedure.”

Point 3: “Section 2.2 last sentence needs an “and” before “RFCA lesions were given””

Response to Point 3:

Thank you very much for your advice. We have therefore amended the manuscript as instructed by adding the following paragraph at “Section 2.2”.

A high-resolution EAM of LA was created with the PENTARAYTM (Biosense Webster) or an Advisor™ HD Grid (Abbott) mapping catheter and RFCA lesions were given for targeting of ablation index 450 or lesion size index 5.0 for pulmonary vein isolation (PVI).”

Point 4: “Abstract: second sentence is a run-on sentence and should be edited.”

Response to Point 4:

Thank you for your very valuable comment. We have therefore amended the manuscript as instructed by adding the following paragraph at abstract section.

“The aim of this study is to identify the novel predictors of effective A-ATP for terminating AF in patients with AF undergoing catheter ablation.”

Point 5: “Section 2.3 Pacing protocol. This needs some more clarity. Presumably the 20 patients with persistent atrial fibrillation were already in atrial fibrillation. But what was the specific protocol for PAF patients presenting in sinus rhythm? How was AF induced? Was isoproterenol used, and what doses? How long was AF allowed to continue before the pacing maneuvers were attempted? An obvious concern is that PAF can terminate on its own, especially if it was triggered by mapping, and it is difficult to tease out whether pacing actually affected the substrate. Can you provide CWT analysis for a control group in whom no pacing was performed but AF spontaneously terminated? Perhaps slowing of cycle length occurs any time AF terminates, and it has nothing to do with the atrial pacing?”

Response to Point 5:

Thank you for your important conceptual observation. If patient’s rhythm was sinus rhythm at the time of procedure, we induced AF by burst pacing from CS (pacing cycle length [CL]: 200ms) on isoproterenol (1~2γ). After AF sustained for more than 30 seconds, we attempted pacing maneuvers.  As you suggested, we cannot confirm whether induced AF terminated on its own or pacing effects, we defined successful AF termination as a termination within 10 seconds after pacing with obvious cycle length changes. Unfortunately, we do not have a data in whom no pacing was performed but AF spontaneously terminated. We have therefore amended the manuscript as instructed by adding the following paragraph at “Section 2.3”.

“If patient’s rhythm was sinus rhythm at the time of procedure, we induced AF by burst pacing from CS (pacing cycle length [CL]: 200ms) on isoproterenol (1~2γ). After AF sustained for more than 30 seconds, we attempted pacing maneuvers. We defined successful AF termination as a termination within 10 seconds after pacing with obvious CL changes.”

Point 6: “Protocol: Please clarify what the difference between decremental pacing and burst pacing is. They appear to be the same thing. Why does burst pacing have a 10ms decrease? Is that between pacing trains? How many pacing trains were given and was there a pause? What was the order of attempts? A, B, C, D, E? Or some random order? Was each patient given all 5 pacing strategies, or only one of the five? Was pacing performed until local capture was demonstrated (e.g., arrows on Figure 1) or was there some other criteria to stop pacing?”

Response to Point 6:

Thank you for your very valuable comments. I am very sorry for my explanation of pacing protocol being difficult to understand. We have therefore amended the manuscript as instructed by adding the following paragraph at “Section 2.3” and a new “Figure 1” according to your questions.

“The pacing protocol used for this study is as follows (Figure 1):

First, we measured mean AF CL for 10 beats. Second, we attempted pacing from HRA using below five pacing CL until AF was terminated (Th1Th2Th3Th4Th5). If AF was not terminated, we changed pacing site to CS distal. If AF was not terminated again, we finally performed simultaneous HRA & CS distal pacing. Pacing study was finished when AF was terminated. Percentage of pacing CL was referred to the protocol of the MINERVA study [2] as follows:

A, Th1. Decremental pacing: 91% of AF CL ×13beats, reduce by 10ms (down to 150ms)

B, Th2. Straight pacing: 84% of AF CL ×13beats, reduce by 10ms (down to 150ms)

C, Th3. Decremental pacing: 81% of AF CL ×13beats, reduce by 10ms (down to 150ms)

D, Th4. Straight pacing: 84% of AF CL ×20beats, reduce by 10ms (down to 150ms)

E, Th5. Decremental pacing: 81% of AF CL ×20beats, reduce by 10ms (down to 150ms)

 Decremental pacing in this study means Pacing CL was reduced by 10ms after each 13/20 pacing beats till 150ms. [ex; AF CL=200ms, Th1; 180ms×13beats + 170ms×13beats + 160ms×13beats + 150ms×13beats]”

Point 7: “Section 2.6 Statistical Analysis: please indicate if P values were one-tailed or two-tailed.”

Response to Point 7:

Thank you for your valuable comment. We used two-tailed P value. We have therefore amended the manuscript as instructed by adding the following paragraph at abstract section.

“A two-tailed P value < 0.05 was considered statistically significant.”

Point 8: “Section 3.1. This needs more data. What were baseline medications of the study group? Clearly, medications such as class I, II, III, or IV antiarrhythmics can influence the study results. Were medications stopped prior to the study? For how long?”

Response to Point 8:

Thank you for highlighting these important comments. We have therefore amended the manuscript as instructed by adding the following paragraph at “Section 3.1” and a new “Table 1”.

“There were no significant differences in age, gender, comorbid conditions, CHADS2 score, medication usage and left ventricular ejection fraction between 2 groups (Table 1).”

“The patients had a mean of 2 ± 1 antiarrhythmic medication, including beta-blocker. Twenty patients were taking anti-arrhythmic drugs (Pilsicainide, n=6; Flecainide, n=7; Bepridil, n=7), and/or twenty-one patients were taking beta blocker (Bisoprolol, n=21). All medications were stopped before the procedure at least 7 days.”

Point 9: “Please also include information about which algorithm (A-E) was used and which one had effect.”

Response to Point 9:

Thank you for your very valuable comment. We have therefore amended the manuscript as instructed by adding the following paragraph at “Section 3.3”

“decremental pacing (A, Th1) was more effective than straight pacing [Successful decremental pacing: 5pt (A, Th1; n=3, C, Th3; n=1, E, Th5; n=1), Successful straight pacing: 2pt (B, Th2; n=1, D, Th4; n=1)].”

Point 10: “Section 4.1: compared to the sinus P wave, the “proximal-distal CS activation” appears to have closely spaced EGMs. The P wave appears very small and narrow. This really has no significance. Is there a CWT to assess here? Also, Figure 6 is confusing. What is the top panel (“RA”) compared to the bottom panel (“CS”)? The text says that there was simultaneous pacing, yet those two tracings appear to have different surface QRSs and appear to be at different time points. Please clarify.”

Response to Point 10:

Thank you for your important conceptual observation. As you suggested, the P wave appears very small and narrow, and the number of captured pacing beats were small, therefore the effects of pacing from CS and HRA were unclear from 12 leads electrograms. However, we can see the change of atrial activation sequences across both the HRA and CS catheters. Unfortunately, recording time was very short, we cannot assess a CWT here. We have therefore amended the manuscript as instructed by adding the following paragraph at “Section 4.1” and a new “Figure 7”.

During simultaneous biatrial straight pacing from the CS and HRA, 1/20 pacing impulses captured the HRA and 4/20 pacing impulses captured the CS (right blue arrows). Because the P wave appears very small and narrow, and the number of captured pacing beats were small, the effects of pacing from CS and HRA were unclear from 12 leads electrograms. However, the atrial activation sequences across both the HRA and CS catheters became reversed after the pacing maneuvers; At both CS and HRA catheters, the activation sequence changed from “distal to proximal” to “proximal to distal” after pacing (red arrows) before termination of AF.”

Point 11: “Section 4.2. A major limitation of this study is that the single site and multi site pacing was performed in sequence. That is, all patients underwent single site first, then if that failed, multi site pacing. We also know that AF can spontaneously initiate and terminate during an AF ablation regardless of pacing. Therefore, a better study would compare two groups of patients, those who had only single site v those who had only multi site; or some sort of cross over where single was done first in half and multi was done first in the other half. Please address this limitation.”

“Section 4.3. There are many more limitations to this study than addressed. See above.”

Response to Point 11:

Thank you for your very valuable comment. As you suggested, this study has a lot of limitations. We have therefore amended the manuscript as instructed by adding the following paragraph at “Section 4.3”. “In addition, a major limitation of this study is that the single site and multi-site pacing was performed in sequence. That is, all patients underwent single site first, then if that failed, multi-site pacing. We also know that AF can spontaneously initiate and terminate during an AF ablation regardless of pacing. Therefore, a better study would compare two groups of patients, those who had only single site vs. those who had only multi-site; or some sort of cross over where single was done first in half and multi was done first in the other half.”

Point 12: “This is a well written and well conducted study that examines atrial pacing during AF that can terminate AF. However, there are a number of minor concerns that should be addressed. More clarity is needed in the protocol for pacing. Figures need to be clarified. The largest major concern is the lack of a control group – many patients would have AF terminated spontaneously despite no pacing. Can the authors provide some information about what CL and CWT analysis shows in that control group? What is the rate of spontaneous termination of AF in those who don’t get atrial pacing?”

Response to Point 12:

Thank you for your very important conceptual observation. As you suggested, the largest major limitation in this study is the lack of a control group. Unfortunately, we do not have a data about CL and CWT analysis in the control group. Furthermore, we do not have a data about the rate of spontaneous termination of AF in those who don’t get atrial pacing. Further study is needed to provide these data. We have therefore amended the manuscript as instructed by adding the following paragraph at “Section 4.3”.

“Finally, the largest major limitation in this study is the lack of a control group - many patients would have AF terminated spontaneously despite no pacing. Further study is needed to provide some information about what CL and CWT analysis shows in that control group and the rate of spontaneous termination of AF in those who don’t get atrial pacing.”

Reviewer 2 Report

In their study Maeda et al. presented their experience regarding simultaneous biatrial atrial anti-tachycardia pacing in 41 carefully selected patients with paroxysmal or persistent AF. The analyzed topic is highly significant from the clinical point of view. The study was properly conducted and the article was written at the appropriate level. Moreover, the raised issue was highlighted using illustrative case descriptions.

I have only several minor suggestions:

  • Please better integrate the aim of the study into the Abstract section.
  • Please clearly emphasize the novelty aspect of the current study.
  • The quality of the deposited figures is insufficient. Please improve it.

Author Response

Dear Reviewer 2

We really appreciate your time and effort for the thorough review of our manuscript. Your comments and questions are very valuable and were quite helpful for improving the manuscript. Please note that the edits and figures have been added in the revised manuscript according to your suggestions. The following is our responses to the questions and comments.

Response to the Comments by Reviewer 2:

Point 1: “Please better integrate the aim of the study into the Abstract section.”

Response to Point 1:

Thank you for your important conceptual observation. We have therefore amended the manuscript as instructed by adding the following paragraph at abstract section.

The aim of this study is to identify the novel predictors of effective A-ATP for terminating AF in patients with AF undergoing catheter ablation.”

Point 2: “Please clearly emphasize the novelty aspect of the current study.”

Response to Point 2: Thank you for your very valuable comment. From the MINERVA study, we know AF can be terminated by A-ATP, however, the mechanism of AF termination is still unknown. And furthermore, there were no study to explore why AF can be terminated by A-ATP. In this study, we sought to evaluate the novel predictors of effective A-ATP for terminating AF in patients with AF undergoing catheter ablation. In addition, this study is the first report to evaluate the predictors of effective A-ATP for terminating AF in patients with AF undergoing catheter ablation. We have therefore amended the manuscript as instructed by adding the following paragraph at “Section 1 and 5”.

“From this study, we know AF can be terminated by A-ATP, however, the mechanism of AF termination is still unknown. And furthermore, there were no study to explore why AF can be terminated by A-ATP. In this study, we sought to evaluate the novel predictors of effective A-ATP for terminating AF in patients with AF undergoing catheter ablation.”

“This study is the first report to evaluate the predictors of effective A-ATP for terminating AF in patients with AF undergoing catheter ablation.”

Point 3: “The quality of the deposited figures is insufficient. Please improve it.”

Response to Point 3: Thank you for your very valuable comment. As you suggested, last 2 figures were very busy. We have therefore added a new “Figure 7”, and we made last figure to separated 2 figures; new “Figure 8” and “Figure 9”.

Round 2

Reviewer 1 Report

The authors have addressed many of my concerns. However, a few remain as below:

Point 2: “Section 2.1: please clarify details of baseline clinical data. How was “PAF” v “PeAF” defined? By ICD10 coding? Over what time period were these patients enrolled? How many PAF patients presented in sinus rhythm and how many PeAF patients presented in sinus rhythm?”

Response to Point 2:

Thank you for highlighting these important comments. Paroxismal AF (PAF) is defined as AF that have been cardioverted during the first 7 days. Persistent AF (PeAF) is defined as AF that persists without interruption for 7 days or longer [3,4]. Furthermore, in patients with PAF, fourteen patients presented in sinus rhythm, whereas no patients presented in sinus rhythm at the time of procedure. We have therefore amended the manuscript as instructed by adding the following paragraph and 2 references at “Section 2.1 and 3.1”.

Paroxismal AF (PAF) is defined as AF that have been cardioverted during the first 7 days. Persistent AF (PeAF) is defined as AF that persists without interruption for 7 days or longer [3,4].”

In patients with PAF, fourteen patients presented in sinus rhythm, whereas no patients presented in sinus rhythm at the time of procedure.”

=> This section remains confusing. Does "cardioversion within 7 days" mean spontaneous cardioversion? Or was some medication or DCCV therapy given? Furthermore, for PAF patients, why did 6 present in AF? Wouldn't that automatically qualify them as having persistent AF? Also, the last sentence is probably missing "PeAF" in the second half (I'm assuming no PeAF patients presented in sinus rhythm).

Point 5: “Section 2.3 Pacing protocol. This needs some more clarity. Presumably the 20 patients with persistent atrial fibrillation were already in atrial fibrillation. But what was the specific protocol for PAF patients presenting in sinus rhythm? How was AF induced? Was isoproterenol used, and what doses? How long was AF allowed to continue before the pacing maneuvers were attempted? An obvious concern is that PAF can terminate on its own, especially if it was triggered by mapping, and it is difficult to tease out whether pacing actually affected the substrate. Can you provide CWT analysis for a control group in whom no pacing was performed but AF spontaneously terminated? Perhaps slowing of cycle length occurs any time AF terminates, and it has nothing to do with the atrial pacing?”

Response to Point 5:

Thank you for your important conceptual observation. If patient’s rhythm was sinus rhythm at the time of procedure, we induced AF by burst pacing from CS (pacing cycle length [CL]: 200ms) on isoproterenol (1~2γ). After AF sustained for more than 30 seconds, we attempted pacing maneuvers.  As you suggested, we cannot confirm whether induced AF terminated on its own or pacing effects, we defined successful AF termination as a termination within 10 seconds after pacing with obvious cycle length changes. Unfortunately, we do not have a data in whom no pacing was performed but AF spontaneously terminated. We have therefore amended the manuscript as instructed by adding the following paragraph at “Section 2.3”.

“If patient’s rhythm was sinus rhythm at the time of procedure, we induced AF by burst pacing from CS (pacing cycle length [CL]: 200ms) on isoproterenol (1~2γ). After AF sustained for more than 30 seconds, we attempted pacing maneuvers. We defined successful AF termination as a termination within 10 seconds after pacing with obvious CL changes.”

=> This is helpful, thanks.

Point 6: “Protocol: Please clarify what the difference between decremental pacing and burst pacing is. They appear to be the same thing. Why does burst pacing have a 10ms decrease? Is that between pacing trains? How many pacing trains were given and was there a pause? What was the order of attempts? A, B, C, D, E? Or some random order? Was each patient given all 5 pacing strategies, or only one of the five? Was pacing performed until local capture was demonstrated (e.g., arrows on Figure 1) or was there some other criteria to stop pacing?”

Response to Point 6:

Thank you for your very valuable comments. I am very sorry for my explanation of pacing protocol being difficult to understand. We have therefore amended the manuscript as instructed by adding the following paragraph at “Section 2.3” and a new “Figure 1” according to your questions.

“The pacing protocol used for this study is as follows (Figure 1):

First, we measured mean AF CL for 10 beats. Second, we attempted pacing from HRA using below five pacing CL until AF was terminated (Th1Th2Th3Th4Th5). If AF was not terminated, we changed pacing site to CS distal. If AF was not terminated again, we finally performed simultaneous HRA & CS distal pacing. Pacing study was finished when AF was terminated. Percentage of pacing CL was referred to the protocol of the MINERVA study [2] as follows:

A, Th1. Decremental pacing: 91% of AF CL ×13beats, reduce by 10ms (down to 150ms)

B, Th2. Straight pacing: 84% of AF CL ×13beats, reduce by 10ms (down to 150ms)

C, Th3. Decremental pacing: 81% of AF CL ×13beats, reduce by 10ms (down to 150ms)

D, Th4. Straight pacing: 84% of AF CL ×20beats, reduce by 10ms (down to 150ms)

E, Th5. Decremental pacing: 81% of AF CL ×20beats, reduce by 10ms (down to 150ms)

 Decremental pacing in this study means Pacing CL was reduced by 10ms after each 13/20 pacing beats till 150ms. [ex; AF CL=200ms, Th1; 180ms×13beats + 170ms×13beats + 160ms×13beats + 150ms×13beats]”

=> I still don't understand what straight pacing means. Why is there a reduction by 10ms in straight pacing for Th2 and Th4? That seems to me like it's decremental.

Point 8: “Section 3.1. This needs more data. What were baseline medications of the study group? Clearly, medications such as class I, II, III, or IV antiarrhythmics can influence the study results. Were medications stopped prior to the study? For how long?”

Response to Point 8:

Thank you for highlighting these important comments. We have therefore amended the manuscript as instructed by adding the following paragraph at “Section 3.1” and a new “Table 1”.

“There were no significant differences in age, gender, comorbid conditions, CHADS2 score, medication usage and left ventricular ejection fraction between 2 groups (Table 1).”

“The patients had a mean of 2 ± 1 antiarrhythmic medication, including beta-blocker. Twenty patients were taking anti-arrhythmic drugs (Pilsicainide, n=6; Flecainide, n=7; Bepridil, n=7), and/or twenty-one patients were taking beta blocker (Bisoprolol, n=21). All medications were stopped before the procedure at least 7 days.”

=> this is a good change.

Point 10: “Section 4.1: compared to the sinus P wave, the “proximal-distal CS activation” appears to have closely spaced EGMs. The P wave appears very small and narrow. This really has no significance. Is there a CWT to assess here? Also, Figure 6 is confusing. What is the top panel (“RA”) compared to the bottom panel (“CS”)? The text says that there was simultaneous pacing, yet those two tracings appear to have different surface QRSs and appear to be at different time points. Please clarify.”

Response to Point 10:

Thank you for your important conceptual observation. As you suggested, the P wave appears very small and narrow, and the number of captured pacing beats were small, therefore the effects of pacing from CS and HRA were unclear from 12 leads electrograms. However, we can see the change of atrial activation sequences across both the HRA and CS catheters. Unfortunately, recording time was very short, we cannot assess a CWT here. We have therefore amended the manuscript as instructed by adding the following paragraph at “Section 4.1” and a new “Figure 7”.

During simultaneous biatrial straight pacing from the CS and HRA, 1/20 pacing impulses captured the HRA and 4/20 pacing impulses captured the CS (right blue arrows). Because the P wave appears very small and narrow, and the number of captured pacing beats were small, the effects of pacing from CS and HRA were unclear from 12 leads electrograms. However, the atrial activation sequences across both the HRA and CS catheters became reversed after the pacing maneuvers; At both CS and HRA catheters, the activation sequence changed from “distal to proximal” to “proximal to distal” after pacing (red arrows) before termination of AF.”

=> this amendment does not address the fact that the top and bottom panels appear to be at different time points -- look at the surface QRSs

Point 11: “Section 4.2. A major limitation of this study is that the single site and multi site pacing was performed in sequence. That is, all patients underwent single site first, then if that failed, multi site pacing. We also know that AF can spontaneously initiate and terminate during an AF ablation regardless of pacing. Therefore, a better study would compare two groups of patients, those who had only single site v those who had only multi site; or some sort of cross over where single was done first in half and multi was done first in the other half. Please address this limitation.”

“Section 4.3. There are many more limitations to this study than addressed. See above.”

Response to Point 11:

Thank you for your very valuable comment. As you suggested, this study has a lot of limitations. We have therefore amended the manuscript as instructed by adding the following paragraph at “Section 4.3”. “In addition, a major limitation of this study is that the single site and multi-site pacing was performed in sequence. That is, all patients underwent single site first, then if that failed, multi-site pacing. We also know that AF can spontaneously initiate and terminate during an AF ablation regardless of pacing. Therefore, a better study would compare two groups of patients, those who had only single site vs. those who had only multi-site; or some sort of cross over where single was done first in half and multi was done first in the other half.”

=> ok

Point 12: “This is a well written and well conducted study that examines atrial pacing during AF that can terminate AF. However, there are a number of minor concerns that should be addressed. More clarity is needed in the protocol for pacing. Figures need to be clarified. The largest major concern is the lack of a control group – many patients would have AF terminated spontaneously despite no pacing. Can the authors provide some information about what CL and CWT analysis shows in that control group? What is the rate of spontaneous termination of AF in those who don’t get atrial pacing?”

Response to Point 12:

Thank you for your very important conceptual observation. As you suggested, the largest major limitation in this study is the lack of a control group. Unfortunately, we do not have a data about CL and CWT analysis in the control group. Furthermore, we do not have a data about the rate of spontaneous termination of AF in those who don’t get atrial pacing. Further study is needed to provide these data. We have therefore amended the manuscript as instructed by adding the following paragraph at “Section 4.3”.

“Finally, the largest major limitation in this study is the lack of a control group - many patients would have AF terminated spontaneously despite no pacing. Further study is needed to provide some information about what CL and CWT analysis shows in that control group and the rate of spontaneous termination of AF in those who don’t get atrial pacing.”

=> ok

Author Response

Dear Sir/Madam

We really appreciate your time and effort for the thorough review of our manuscript. Your comments and questions are very valuable and were quite helpful for improving the manuscript. Please note that the edits and figures have been added in the revised manuscript according to your suggestions. The following is our responses to the questions and comments.

Response to the Comments by Reviewer 1:

Point 2: “This section remains confusing. Does "cardioversion within 7 days" mean spontaneous cardioversion? Or was some medication or DCCV therapy given? Furthermore, for PAF patients, why did 6 present in AF? Wouldn't that automatically qualify them as having persistent AF? Also, the last sentence is probably missing "PeAF" in the second half (I'm assuming no PeAF patients presented in sinus rhythm).”

Response to Point 2:

Thank you for your important conceptual observation. According to ESC and AHA/ACC/HRS guideline, paroxismal AF (PAF) is defined as AF that terminates spontaneously or with intervention within 7 days of onset. And episodes may recur with variable frequency. Persistent AF (PeAF) is defined as continuous AF that is sustained > 7 days. [3,4]. Furthermore, we are very sorry for my mistakes. We were confused. In patients with PAF, all patients presented in sinus rhythm at the time of procedure. We have therefore amended the manuscript as instructed by adding the following paragraph at “Section 2.1 and 3.1”.

According to the ESC and AHA/ACC/HRS guidelines, paroxismal AF (PAF) is defined as AF that terminates spontaneously or with intervention within 7 days of onset. And episodes may recur with variable frequency. Persistent AF (PeAF) is defined as continuous AF that is sustained > 7 days. [3,4].”

“In patients with PAF, all patients presented in sinus rhythm at the time of procedure.”

Point 6: “I still don't understand what straight pacing means. Why is there a reduction by 10ms in straight pacing for Th2 and Th4? That seems to me like it's decremental.”

Response to Point 6:

Thank you for your very valuable comments. We are very sorry for our explanation of pacing protocol being difficult to understand. Decremental pacing in this study means “Pacing CL is reduced by 10ms continuously after each 13/20 pacing beats till 150ms.”. On the other hand, straight pacing in this study means “If first pacing fails to terminate AF, pacing CL is reduced by 10ms and pace again. If AF was not terminated again, pacing CL is reduced by 10ms and pace again. Pacing is repeated till 150ms or AF termination.”. We have therefore amended the manuscript as instructed by adding the following paragraph at “Section 2.3” and a new “Figure 1 edit” according to your questions.

“A, Th1. Decremental pacing: 91% of AF CL ×13beats, reduce by 10ms continuously (down to 150ms).

B, Th2. Straight pacing: 84% of AF CL ×13beats. If first pacing fails to terminate AF, pacing CL is reduced by 10ms and pace again. If AF was not terminated again, pacing CL is reduced by 10ms and pace again. Pacing is repeated till 150ms or AF termination.

C, Th3. Decremental pacing: 81% of AF CL ×13beats, reduce by 10ms continuously (down to 150ms).

D, Th4. Straight pacing: 84% of AF CL ×20beats. If first pacing fails to terminate AF, pacing CL is reduced by 10ms and pace again. If AF was not terminated again, pacing CL is reduced by 10ms and pace again. Pacing is repeated till 150ms or AF termination.

E, Th5. Decremental pacing: 81% of AF CL ×20beats, reduce by 10ms continuously (down to 150ms).

Decremental pacing in this study means Pacing CL is reduced by 10ms continuously after each 13/20 pacing beats till 150ms. [ex; AF CL=200ms, Th1; 180ms×13beats170ms×13beats160ms×13beats150ms×13beats continuously.] Straight pacing in this study means If first pacing fails to terminate AF, pacing CL is reduced by 10ms and pace again. If AF was not terminated again, pacing CL is reduced by 10ms and pace again. Pacing is repeated till 150ms or AF termination.. [ex; AF CL=200ms, Th2; 170ms×13beatsif not terminated, 160ms×13beatsif not terminated, 150ms×13beats.].”

Point 10: “this amendment does not address the fact that the top and bottom panels appear to be at different time points -- look at the surface QRSs”

Response to Point 10:

Thank you for your valuable comment. We are very sorry for the indistinct reduced image. Top panel and bottom panels were recorded at the same time.  To recognize the timing of top and bottom panels, we drew the blue dotted lines between QRS and V sequence of CS catheter in “Figure 7 edit”. We have therefore amended the manuscript as instructed by adding the following paragraph at “figure legends of Figure 7”

Blue dotted lines show the timing of QRS.”

The below panel is the enlarged view at this point.

(We are sorry we can not update a panel in this page. Please refer to the word file "Response to JCM 2")

We believe that the revised manuscript has addressed all the comments brought up by the reviewers and as a result, the manuscript is much improved. We sincerely hope that this version can be considered for publication in Journal of Clinical Medicine. Thank you in advance for your time.

Very sincerely yours

Shingo Maeda, MD, PhD

Arrhythmia Advanced Therapy Center,

AOI Universal Hospital